# LoveDA: A Remote Sensing Land-Cover Dataset for Domain Adaptation Semantic Segmentation

**Junjue Wang, Zhuo Zheng, Ailong Ma, Xiaoyan Lu, Yanfei Zhong**
State Key Laboratory of Information Engineering in Surveying, Mapping, and Remote Sensing
Wuhan University, Wuhan 430074, China
{kingdrone,zhengzhuo,maailong007,luxiaoyan,zhongyanfei}@whu.edu.cn

## Abstract

Deep learning approaches have shown promising results in remote sensing high spatial resolution (HSR) land-cover mapping. However, urban and rural scenes can show completely different geographical landscapes, and the inadequate generalizability of these algorithms hinders city-level or national-level mapping. Most of the existing HSR land-cover datasets only focus on improvement of the semantic segmentation in one domain (urban or rural), thereby ignoring the model transferability. In this paper, we introduce the Land-cOVEr Domain Adaptation semantic segmentation (LoveDA) dataset to promote large-scale land-cover mapping. The LoveDA dataset contains 3338 aerial images with 86516 annotated objects for seven common land-cover categories. Compared to the existing datasets, the LoveDA dataset encompasses two domains (urban and rural), which brings considerable challenges due to the: 1) multi-scale objects; 2) complex background samples; and 3) inconsistent class distributions. The LoveDA dataset is suitable for both land-cover semantic segmentation and unsupervised domain adaptation (UDA) tasks. Accordingly, we benchmarked the LoveDA dataset on nine semantic segmentation methods and eight UDA methods. Some exploratory studies were also carried out to find alternative ways to address these challenges. The code and data will be available at: https://github.com/Junjue-Wang/LoveDA

## 1 Introduction

With the continuous development of society and economy, the human living environment is gradually being differentiated, and can be divided into urban and rural zones [7]. High spatial resolution (HSR) remote sensing technology can help us to better understand the geographical and ecological environment. Specifically, land-cover semantic segmentation in remote sensing is aimed at determining the land-cover type at every image pixel. The existing HSR land-cover datasets such as the Gaofen Image Dataset (GID) [34], DeepGlobe [8], Zeebruges [22], and Zurich Summer [37] contain large-scale images with pixel-wise annotations, thus promoting the development of fully convolutional networks (FCNs) in the field of remote sensing [6]. However, these datasets are designed for single-domain semantic segmentation, and they ignore the diverse styles among geographic areas. For urban and rural areas, in particular, the manifestation of the land cover is completely different, in the class distributions, object scales, and pixel spectra. In order to improve the model generalizability for large-scale land-cover mapping, appropriate datasets are required.

In this paper, we introduce an HSR dataset for Land-cOVEr Domain Adaptation semantic segmentation (LoveDA) for use in two challenging tasks: semantic segmentation and UDA. Compared with the UDA datasets [21, 35] that using simulated images, the LoveDA dataset contains real urban and rural remote sensing images. Exploring the use of deep transfer learning methods on this dataset will be a meaningful way to promote large-scale land-cover mapping. The major characteristics of

Submitted to the 35th Conference on Neural Information Processing Systems (NeurIPS 2021) Track on Datasets and Benchmarks. Do not distribute.

this dataset are summarized as follows: **1) Multi-scale objects.** The HSR images were collected from 10 complex urban and rural scenes, covering 11 administrative districts in China. The objects in the same category are in completely different geographical landscapes in the different scenes, which increases the scale variation. **2) Complex background samples.** The remote sensing semantic segmentation task is always faced with the complex background samples (i.e., land-cover objects that are not of interest) [27, 47], which is particularly the case in the LoveDA dataset. The high-resolution and different complex scenes bring more rich details as well as larger intra-class variance for the background samples. **3) Inconsistent class distributions.** The urban and rural scenes have different class distributions. The urban scenes with high population densities contain lots of artificial objects such as buildings and roads. In contrast, the rural scenes include more natural elements, such as forest and water. The inconsistent class distributions pose a special challenge for the UDA task.

As the LoveDA dataset was built with two tasks in mind, both advanced semantic segmentation and UDA methods were evaluated. Several exploratory experiments were also conducted to solve the particular challenges inherent in this dataset, and to inspire further research. A stronger representational architecture and UDA method are needed to jointly promote large-scale land cover mapping.

## 2 Related Work

### 2.1 Land-cover semantic segmentation datasets

Land-cover semantic segmentation, as a long-standing research topic, has been widely explored over the past decades. The early research relied on low- and medium-resolution datasets, such as MCD12Q1 [30], the National Land Cover Database (NLCD) [11], GlobeLand30 [12], LandCoverNet [1], etc. However, these studies all focused on large-scale mapping and analysis from a macro-level. With the advancement of remote sensing technology, massive HSR images are now being obtained on a daily basis from both spaceborne and airborne platforms. Due to the advantages of the clear geometrical structure and fine texture, HSR land-cover datasets are tailored for specific scenes at a micro-level. As is shown in Table 1, datasets such as ISPRS Potsdam [1], ISPRS Vaihingen [2], Zurich Summer [37], and Zeebruges [22] are designed for urban parsing. These datasets only contain a small number of annotated images, pixels, and instances. In contrast, DeepGlobe [8] and LandCover.ai [2] focus on rural areas with a larger scale, in which the homogeneous areas contain few man-made structures. The GID dataset[34] was collected from different cities in China, covering both urban areas and the surrounding rural areas. Although the LandCoverNet and GID datasets contain both urban and rural areas, the geo-locations of these released images are private. Therefore, the urban and rural areas are not able to be divided. In addition, the identifications of cities in released GID images have been already removed so it is hard to perform UDA tasks. Considering limited coverage and annotation cost, the existing HSR datasets only focus on improvement of the semantic segmentation in one domain (urban or rural).

Table 1: Comparison between LoveDA and the main land-cover semantic segmentation datasets.

| Dataset | Sensor | Area (km$^2$) | Resolution (m) | Classes | Image width | Images | Domain | | Task | |
|---|---|---|---|---|---|---|---|---|---|---|
| | | | | | | | Urban | Rural | SS | UDA |
| LandCoverNet [1] | Sentinel-2 | 30000 | 10 | 7 | 256 | 1980 | ✓ | ✓ | ✓ | |
| GID [34] | GF-2 | 75900 | 4 | 5 | 4800∼6300 | 150 | ✓ | ✓ | ✓ | |
| LandCover.ai [2] | Airborne | 216.27 | 0.25∼0.5 | 3 | 4200∼9500 | 41 | | ✓ | ✓ | |
| Zurich Summer [37] | QuickBird | 9.37 | 0.6 | 8 | 622∼1830 | 20 | ✓ | | ✓ | |
| DeepGlobe [8] | WorldView-2 | 1716.9 | 0.5 | 7 | 2448 | 1146 | | ✓ | ✓ | |
| Zeebruges [22] | Airborne | 1.75 | 0.05 | 8 | 10000 | 7 | ✓ | | ✓ | |
| ISPRS Potsdam [1] | Airborne | 3.42 | 0.05 | 6 | 6000 | 38 | ✓ | | ✓ | |
| ISPRS Vaihingen [2] | Airborne | 1.38 | 0.09 | 6 | 1887∼3816 | 33 | ✓ | | ✓ | |
| LoveDA (Ours) | Airborne | 300.48 | 0.3 | 7 | 1024 | 3338 | ✓ | ✓ | ✓ | ✓ |

The abbreviations are: SS – semantic segmentation, UDA – unsupervised domain adaptation.

These HSR land-cover datasets have all promoted the development of semantic segmentation, and many variants of FCNs [18] have been evaluated [6, 13, 25, 40]. Recently, some UDA methods have been developed from the combination of two public datasets [43]. However, directly utilizing

---

[1]http://www2.isprs.org/commissions/comm3/wg4/2d-sem-label-potsdam.html

[2]http://www2.isprs.org/commissions/comm3/wg4/2d-sem-label-vaihingen.html

combined datasets may result in two problems: **1) Insufficient shared categories.** Different datasets are designed for different purposes, and the insufficient shared categories limit further exploration. **2) Inconsistent annotation granularity.** The different spatial resolutions and labeling styles lead to different annotation granularities, which can result in unreliable conclusions. Compared with these datasets, LoveDA dataset encompasses two domains (urban and rural), representing a novel UDA task for land-cover mapping. The LoveDA dataset also has the following advantages in statistical diversity: **1.Considerable geographic area:** As is shown in the Table 1, the area of LoveDA dataset surpasses all existing airborne datasets and demonstrates its diversity. **2.Sub-meter resolution:** Compared with GID and LandCoverNet datasets which cover larger scale areas due to lower spatial resolutions, our spatial details are more than ten times richer than them. The rich feature details increase our diversity **3.Fine annotations:** The LoveDA dataset has instance-level annotations compared with the DeepGlobe dataset. The fine annotation granularity increases the diversity of samples, i.e. every building has its unique shape (Figure 1). **4.Complex scenes:** The LoveDA dataset was constructed from both urban and rural scenes, further reducing the biased statistics. In addition, the area of urban scenes ($\approx 150\ km^2$) far exceeds the existing urban datasets, which can also highlight its value and significance in urban mapping.

## 2.2 Unsupervised domain adaptation

UDA is aimed at transferring a model trained on the source domain to the target domain. Some conventional image classification studies [19, 31, 36] have directly minimized the discrepancy of the feature distributions to extract domain-invariant features. The recent works have mainly proceeded in two directions, i.e., adversarial training and self-training.

**Adversarial training**. In adversarial training, the architecture includes a feature extractor and a discriminator. The extractor aims to learn domain-invariant features, while the discriminator attempts to distinguish these features. For semantic segmentation, Tsai et al. [35] considered the semantic outputs containing spatial similarities between the different domains, and adapted the structured output space for segmentation (AdaptSeg) with adversarial learning. Luo et al. [21] introduced a category-level adversarial network (CLAN) to align each class with an adaptive adversarial loss. Differing from the binary discriminators, Wang et al. [38] proposed a fine-grained adversarial learning framework for domain adaptive semantic segmentation (FADA), aligning the class-level features. From the aspect of structure, the transferable normalization (TransNorm) method [41] was proposed to enhance the transferability of the FCN-based feature extractors. All these advanced adversarial learning methods were implemented on the LoveDA dataset for evaluation.

**Self-training**. Self-training involves alternately generating pseudo-labels on the target data and fine-tuning the model. Recently, the self-training UDA methods have focused on improving the quality of the pseudo-labels [44, 50]. Zou et al. [49] proposed a class-balanced self-training (CBST) strategy to sample pseudo-labels, thus avoiding the dominance of the large classes. Mei et al. [23] used an instance adaptive self-training (IAST) selector for sample balance. Lian et al. [16] designed the self-motivated pyramid curriculum (PyCDA) to observe the target properties, and fused multi-scale features. In addition to testing these self-training methods on the LoveDA dataset, we also performed the multi-scale analysis for the PyCDA.

**UDA in the remote sensing community**. The early UDA methods focused on scene classification tasks [20, 26]. Recently, adversarial training [10, 32] and self-training [34] have been studied for UDA land-cover semantic segmentation . The main algorithms follow the general UDA approach in the computer vision field, with some improvements. However, with only the public datasets, the advancement of the UDA algorithms has been limited by the insufficient shared categories and the inconsistent annotation granularity. Hence, we built the LoveDA dataset to provide a more challenging and solid platform for UDA in remote sensing.

## 3 Dataset Description

China has been experiencing a rapid process of urbanization since the implementation of the "reform and opening up" policy in 1978 [17]. The city of Nanjing, which is regarded as an important national research center and transportation junction, is the epitome of the developed cities in China. Therefore, the LoveDA dataset was constructed using 0.3 m aerial images obtained from Nanjing in July 2016, covering 300.48km$^2$ (Figure 1).

## 3.1 Image Distribution and Division

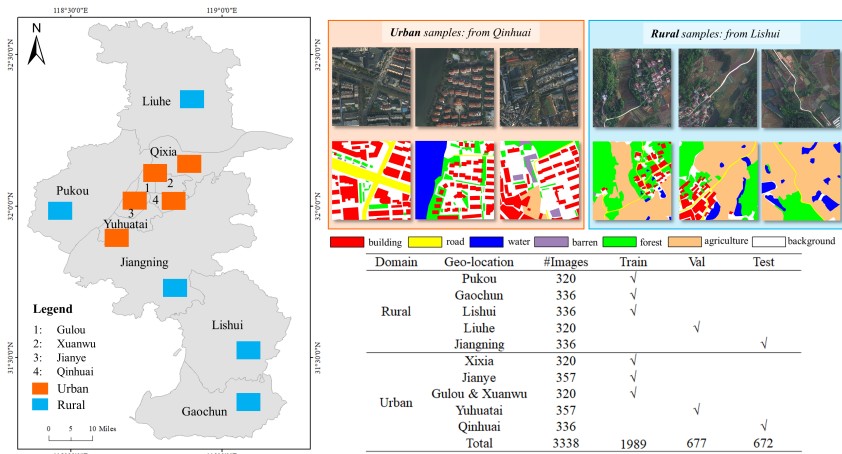

| Domain | Geo-location | #Images | Train | Val | Test |
|--------|--------------|---------|-------|-----|------|
| Rural | Pukou | 320 | √ | | |
| | Gaochun | 336 | √ | | |
| | Lishui | 336 | √ | | |
| | Liuhe | 320 | | √ | |
| | Jiangning | 336 | | | √ |
| Urban | Xixia | 320 | √ | | |
| | Jianye | 357 | √ | | |
| | Gulou & Xuanwu | 320 | √ | | |
| | Yuhuatai | 357 | | √ | |
| | Qinhuai | 336 | | | √ |
| | Total | 3338 | 1989 | 677 | 672 |

Figure 1: Overview of the dataset distribution. The images were collected from 10 spatially independent areas, covering 11 administrative districts in Nanjing. The examples were sampled from the Qinhuai (urban) and Lishui (rural) areas.

Data from the rural and urban areas were collected referring to the "Urban and Rural Division Code" issued by the National Bureau of Statistics. There are six districts (Gulou, Xuanwu, Jianye, Qinhuai, Qixia, and Yuhuatai) in the center of Nanjing, which are all densely populated ($> 1000 \mathrm{~person/km}^2$) [45]. As shown in Figure 1, we selected five economically developed areas as representative urban areas: Qixia, Gulou & Xuanwu, Jianye, Qinhuai, and Yuhuatai. The other five areas were selected as rural areas with a low population density, i.e., Liuhe, Pukou, Jiangning, Lishui, and Gaochun. All the HSR images were captured with a Leica DMC digital camera mounted on an airborne platform. The spatial resolution is 0.3 m, with red, green, and blue bands. After geometric registration and pre-processing, each area is covered by $1024 \times 1024$ images, without overlap. Considering Tobler's First Law, i.e., everything is related to everything else, but near things are more related than distant things [33], the training, validation, and test sets were split so that they were spatially independent (Figure 1), thus enhancing the difference between the split sets. There are two tasks that can be evaluated on the LoveDA dataset: **1) Semantic segmentation**. There are 1989 images from six areas for training, and the others are for validation and testing. The training, validation, and test sets cover both urban and rural areas. **2) Unsupervised domain adaptation**. The UDA process considers two cross-domain adaptation sub-tasks: *a) Urban → Rural*. The images from the Qixia, Jianye, and Gulou & Xuanwu areas are included in the source training set. The images from Liuhe are included in the validation set, and the Jiangning images included in the test set. The *Oracle* setting is designed to test the upper limit of accuracy in a single domain [28]. Hence, the training images were collected from the Pukou, Gaochun, and Lishui areas. *b) Rural → Urban*. The images from the Pukou, Gaochun and Lishui areas are included in the source training set. The images from Yuhuatai are used for the validation set, and the Qinhuai images are used for the test set. In the *Oracle* setting, the training images cover the Qixia, Jianye, and Gulou & Xuanwu areas.

With the division of these images, a comprehensive annotation pipeline was adopted, including professional annotators and strict inspection procedures [42]. Further details of the annotation can be found in the Appendix. The seven common land-cover types were considered, i.e., buildings, road, water, forest, agriculture, and background classes.

## 3.2 Statistics for LoveDA

Some statistics of the LoveDA dataset are analyzed in this section. With the collection of public HSR land-cover datasets, the number of labeled objects and pixels has been counted. As is shown in the Figure 2(a), the DeepGlobe dataset contains the largest number of labeled pixels ($\approx 4.8$ billion) and covers a large-scale rural area. Our proposed LoveDA dataset contains 86516 annotated objects of seven categories. This is because the LoveDA dataset covers large-scale urban scenes (five areas of about $151.05 \mathrm{~km}^2$), which contain many buildings (Figure 2(b)). Among the artificial objects,

the number of road objects is small due to the continuous characteristic of roads. There are a lot of objects in the forest class because the trees in the urban scenes are scattered. As is shown in Figure 2(c), the background class contains the most pixels with complex samples [27, 47]. The **complex background samples** have larger intra-class variance in the complex scenes and cause serious false alarms.

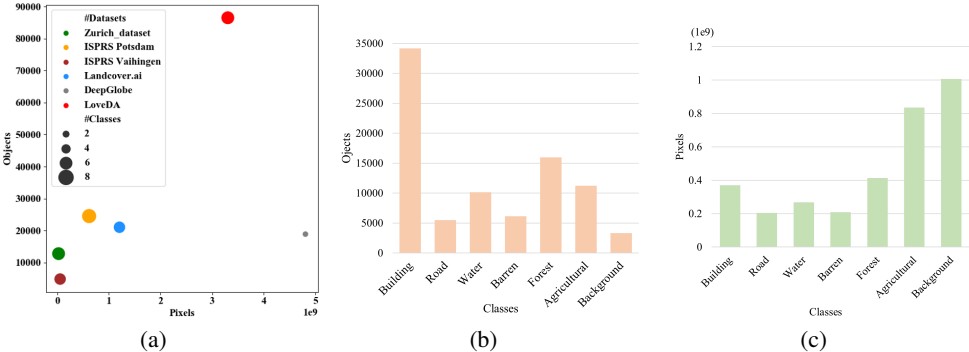

(a)  (b)  (c)

Figure 2: Statistics for the pixels and objects in the LoveDA dataset. (a) Number of objects vs. number of pixels. The radius of the circles represents the number of classes. (b) Histogram of the number of objects for each class. (c) Histogram of the number of pixels for each class.

167

## 3.3 Differences Between Urban and Rural Scenes

During the process of urbanization, cities differentiate into rural and urban forms. Affected by different lifestyles, the living environment also presents different styles, especially for land cover. In this section, we list the main differences between the urban and rural scenes, which reveal the meaning and challenges of the UDA task. For the LoveDA dataset, the main differences come from the shape, layout, scale, spectra, and class distribution. As is shown in Figure 1, the buildings in the urban area are neatly arranged, with various shapes, while the buildings in the rural area are disordered, with simple shapes. The roads are wide in the urban scenes. In contrast, the roads are narrow in the rural scenes. Water is often present in the form of large-scale rivers or lakes in the urban scenes, while small-scale ponds and ditches are common in the rural scenes. The agricultural land is found in the gaps between the houses in the urban scenes, but occurs in a large-scale and continuously distributed form in the rural scenes.

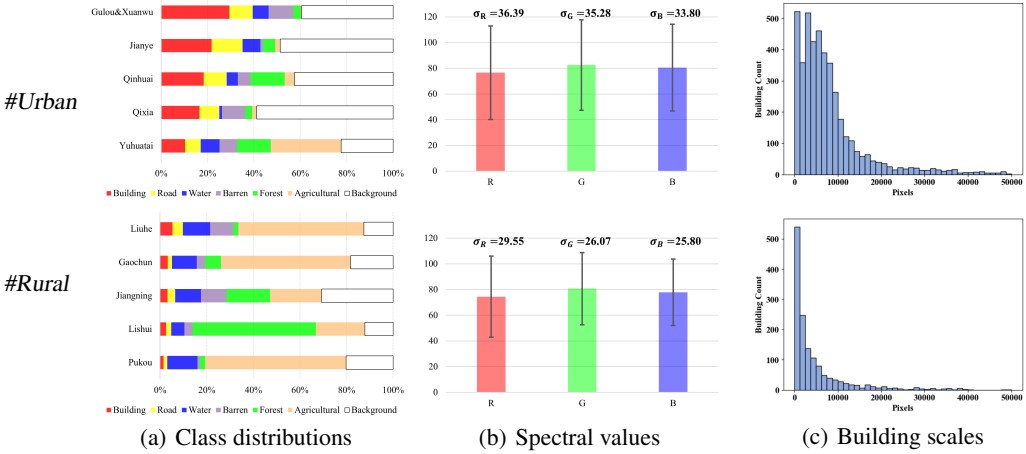

(a) Class distributions  (b) Spectral values  (c) Building scales

Figure 3: Statistics for the urban and rural scenes in LoveDA. (a) Class distribution. (b) Spectral statistics. The mean and standard deviation ($\sigma$) for 10 areas are reported. (c) Distribution of the building sizes. The Jianye (urban) and Lishui (rural) scenes are reported.

For the class distribution, spectra, and scale, the related statistics are reported in Figure 3. The urban areas always contain more man-made objects such as buildings and roads due to their high population density (Figure 3(a)). In contrast, the rural areas have more agricultural land. The **inconsistent class distributions** between the urban and rural scenes increases the difficulty of model generalization. For the spectral statistics, the mean values are similar (Figure 3(b)). Because of the large-scale homogeneous geographical areas, such as agriculture, forest and water, the rural images have lower standard deviations. This reflects the fact that the urban features are more complex than those in the rural scenes. As is shown in Figure 3(c), most of the buildings have relatively small scales in the rural areas, representing the "long tail" phenomenon. However, the buildings in the urban scenes have a larger size variance. Scale differences also exist in the other categories, as shown in Figure 1. The **multi-scale objects** require the models to have multi-scale capture capabilities. When faced with large-scale land cover mapping tasks, the differences between urban and rural scenes bring new challenges to the model transferability.

# 4 Experiments

## 4.1 Semantic Segmentation

For the semantic segmentation task, the general architectures as well as their variants, and particularly those most often used in remote sensing, were tested on the LoveDA dataset. Specifically, the selected networks were: UNet[29], UNet++[48], LinkNet[3], DeepLabV3+[5], PSPNet[46], FCN8S[18], PAN[15], Semantic-FPN[14], HRNet[39], and FarSeg[47]. Following the common practice[18, 39], we use the intersection over union (IoU) to report the semantic segmentation accuracy. With respect to the IoU for each class, the mIoU represents the mean of the IoUs over all the categories. Besides, the prediction speed is reported with $512 \times 512$ inputs, using frames per second (FPS).

Table 2: Semantic segmentation results obtained on the test set of LoveDA.

| Method | Backbone | IoU per category (%) | | | | | | | mIoU (%) | Speed (FPS) |
|---|---|---|---|---|---|---|---|---|---|---|
| | | Background | Building | Road | Water | Barren | Forest | Agriculture | | |
| FCN8S [18] | VGG16 | 48.28 | 51.34 | 50.16 | 70.2 | 17.93 | 47.49 | 63.69 | 49.87 | 86.02 |
| DeepLabV3+ [5] | ResNet50 | 46.96 | 51.88 | 53.01 | 72.85 | 14.56 | 45.18 | 65.11 | 49.94 | 71.35 |
| PSPNet [46] | ResNet50 | 49.09 | 54.41 | 53.3 | 72.86 | 11.14 | 47.34 | 66.09 | 50.61 | 27.22 |
| UNet [29] | ResNet50 | 48.89 | 56.31 | 51.82 | 71.86 | 15.04 | 45.57 | 65.25 | 50.68 | 75.33 |
| UNet++ [48] | ResNet50 | 48.75 | 55.3 | 52.61 | 73.01 | 14.06 | 48.05 | 68.37 | 51.45 | 61.09 |
| PAN [15] | ResNet50 | 48.48 | 55.13 | 51.83 | 70.73 | 16.89 | 46.40 | 65.37 | 50.69 | 73.98 |
| Semantic-FPN [14] | ResNet50 | 48.23 | 51.92 | 54.78 | 71.36 | **21.41** | 46.09 | 67.08 | 51.55 | 25.5 |
| LinkNet [3] | ResNet50 | 48.56 | 53.69 | 52.76 | 73.02 | 16.37 | 47.76 | 66.54 | 51.24 | 67.01 |
| FarSeg [47] | ResNet50 | 49.42 | 55.23 | 53.89 | 72 | 12.55 | 47.91 | 65.41 | 50.92 | 66.99 |
| HRNet [39] | W32 | 50.48 | 56.55 | 54.33 | 73.72 | 19 | 49.99 | 69.53 | 53.37 | 16.74 |

**Implementation details**. The data splits followed the table in Figure 1. During the training, we used the Stochastic Gradient Descent (SGD) optimizer with a momentum of $0.9$ and a weight decay of $10^{-4}$. The learning rate was initially set to $0.01$, and a 'poly' schedule with power $0.9$ was applied. The number of training iterations was set to $15k$ with a batch size of 12. For the data augmentation, $512 \times 512$ patches were randomly cropped from the raw images, with random mirroring and rotation. The backbones used in all the networks were pre-trained on ImageNet.

**Multi-scale objects**. As ground objects show considerable scale variance, especially in complex scenes (§3.3), a powerful multi-scale feature fusion ability is required. There are three noticeable observations from Table 2: 1) UNet++ outperforms UNet due to its nested cross-scale connections between different scales. 2) Among the different fusion strategies, UNet++, Semantic-FPN, LinkNet and HRNet outperform DeepLabV3+ and PSPNet. This demonstrates that the multi-scale fusion between different layers works better than the in-module fusion. 3) HRNet outperforms the other methods, due to its sophisticated architecture, where the features are repeatedly exchanged across different scales. 4) As is shown in Table 3, multi-scale augmentation (with scale = {0.5, 0.75, 1.0, 1.25, 1.5, 1.75}) was conducted during the training and testing, further improving the performance.

**Complex background samples**. The complex background samples in LoveDA dataset cause serious false alarms in HRS imagery semantic segmentation [9, 47]. As is shown in Figure 4, the four confusion matrices show that lots of objects were misclassified into background. This observation is consistent with our analysis in §3.2, so that we adopted an additional loss for the background

Table 3: Multi-scale augmentation on different methods.

| Method | Backbone | mIoU (%) | | |
| --- | --- | --- | --- | --- |
| | | Baseline | +MST | +MSTT |
| Semantic-FPN | ResNet50 | 51.55 | 51.71 | 52.01 |
| UNet | ResNet50 | 50.68 | 51.21 | 51.93 |
| DeepLabV3+ | ResNet50 | 49.94 | 50.03 | 50.6 |
| HRNet | W32 | 53.37 | 54.09 | 54.32 |

The abbreviations are: MST – multi-scale augmentation during training. MSTT – multi-scale augmentation during training and testing.

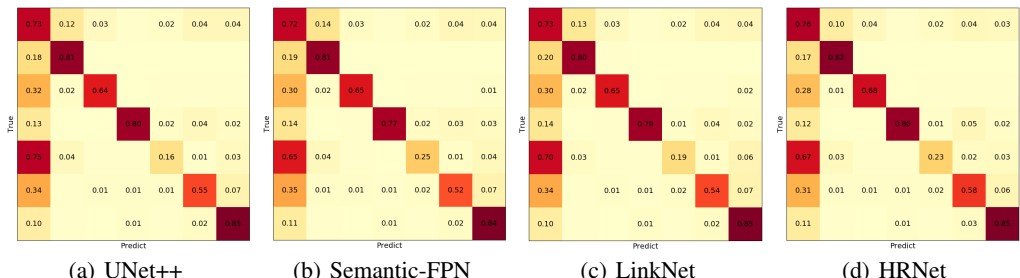

| (a) UNet++ | (b) Semantic-FPN | (c) LinkNet | (d) HRNet |

Figure 4: The confusion matrices for the test set. The categories from left to right (up to down): background, building, road, water, barren, forest, agriculture.

supervision. Dice loss [24] and binary cross-entropy loss were utilized with the corresponding modulation factors. We calculated the total loss as: $L_{total} = L_{ce} + \alpha L_{dice} + \beta L_{bce}$, where $L_{ce}$ denotes the original cross-entropy loss. Table 4 and Table 5 additionally report the precision (P), recall (R) and F1-score (F1) of the background class with varying modulation factors. Besides, the standard deviations are also reported after 3 runs. Table 4 shows that the addition of dice loss improves the background accuracy and the overall performance. The combination of dice loss and binary cross-entropy loss performs well because they optimize the background class from different directions.

Table 4: Varied $\alpha$ for the dice loss in HRNet

| $\alpha$ | Background | | | mIoU (%) |
| --- | --- | --- | --- | --- |
| | P (%) | R (%) | F1(%) | |
| 0 | 61.64 | 73.59 | 67.09 | $53.37 \pm 0.16$ |
| 0.1 | 61.65 | 75.07 | 67.70 | $53.57 \pm 0.12$ |
| 0.2 | 61.94 | 76.39 | 68.41 | $53.97 \pm 0.19$ |
| 0.5 | 62.33 | 75.90 | 68.45 | $54.16 \pm 0.19$ |
| 0.7 | 62.21 | 76.38 | 68.57 | $\mathbf{54.35} \pm 0.15$ |
| 1.0 | 62.25 | 76.84 | 68.78 | $54.26 \pm 0.11$ |
| 1.5 | 61.65 | 75.07 | 67.70 | $53.49 \pm 0.16$ |

Table 5: Varied $\beta$ for the binary cross-entropy loss in HRNet (w. optimal $\alpha$)

| $\beta$ | $\alpha$ | Background | | | mIoU (%) |
| --- | --- | --- | --- | --- | --- |
| | | P (%) | R (%) | F1(%) | |
| 0 | 0 | 61.64 | 73.59 | 67.09 | $53.37 \pm 0.17$ |
| 0.1 | 1.0 | 62.59 | 75.20 | 68.32 | $54.28 \pm 0.16$ |
| 0.2 | 1.0 | 62.51 | 75.48 | 68.38 | $54.13 \pm 0.13$ |
| 0.5 | 1.0 | 62.54 | 72.39 | 67.11 | $53.57 \pm 0.10$ |
| 0.5 | 0.7 | 62.96 | 76.14 | 68.93 | $\mathbf{54.94} \pm 0.08$ |
| 0.7 | 0.7 | 62.75 | 73.69 | 67.78 | $54.45 \pm 0.15$ |
| 1.0 | 0.7 | 62.42 | 73.76 | 67.62 | $53.85 \pm 0.07$ |

**Visualization**. Some representative results are shown in Figure 5. With the shallow backbone (VGG16), FCN8S can hardly recognize the road due to its lack of feature extraction capability. The other methods which utilize deep layers can produce better results. Because of the disorderly arrangement and varied scales, the edges of the buildings are hard to extract accurately, and the small buildings are easy to miss. In contrast, the natural classes, especially water, achieve higher accuracies for all the methods. This may be because natural objects have strong spectral homogeneity and low intra-class variance [34]. The forest is easy to misclassify into agriculture because these classes have similar spectra. Because of the high-resolution retention and multi-scale fusion, HRNet produces the best visualization result, especially in the details.

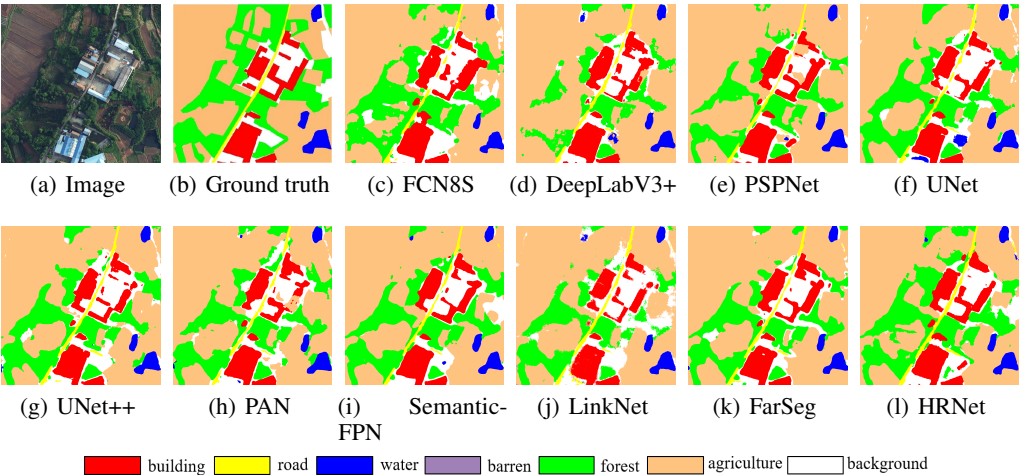

(a) Image    (b) Ground truth    (c) FCN8S    (d) DeepLabV3+    (e) PSPNet    (f) UNet

(g) UNet++    (h) PAN    (i) Semantic-FPN    (j) LinkNet    (k) FarSeg    (l) HRNet

🟥 building    🟨 road    🟦 water    🟪 barren    🟩 forest    🟧 agriculture    ⬜ background

Figure 5: Visual results on images from the LoveDA test set in the Liuhe (**Rural**) area. The artificial classes (building and road) obtain lower performances than the natural classes (water, agricultural). The forest and agricultural classes are easy to misclassify due to their similar spectra.

## 4.2 Unsupervised Domain Adaptation

The advanced UDA methods were evaluated on the LoveDA dataset. In addition to the original metric-based approach of MCD [36], two mainstream UDA approaches were tested, i.e., adversarial training (AdaptSeg [35], CLAN [21], TransNorm [41], FADA [38]) and self-training (CBST [49], PyCDA [16], IAST [23]).

Table 6: Unsupervised domain adaptation results obtained on the test set of the LoveDA dataset.

| Domain | Method | Type | IoU (%) | | | | | | | mIoU(%) |
|---|---|---|---|---|---|---|---|---|---|---|
| | | | Background | Building | Road | Water | Barren | Forest | Agriculture | |
| | Oracle | - | 46.73 | 47.07 | 33.32 | 65.57 | 11.65 | 54.33 | 57.66 | 45.19 |
| **Urban** ↓ Rural | Source only | - | 32.85 | 25.49 | **29.23** | 52.17 | 9.69 | 33.29 | 32.99 | 30.82 |
| | MCD [36] | - | 34.71 | 28.71 | 27.98 | 46.49 | 23.57 | 49.95 | 26.92 | 34.05 |
| | AdaptSeg [35] | AT | 33.85 | 29.49 | 20.71 | 45.71 | 26.03 | 48.85 | 30.98 | 33.66 |
| | CLAN [21] | AT | **36.61** | 32.29 | 22.12 | 41.58 | 31.42 | 43.15 | 36.08 | 34.74 |
| | TransNorm [41] | AT | 28.15 | 22.35 | 19.26 | 34.94 | 0.56 | 13.57 | 0.37 | 18.97 |
| | FADA [38] | AT | 34.30 | 29.37 | 18.41 | 48.57 | 36.68 | 45.43 | 32.29 | 35.45 |
| | CBST [49] | ST | 32.89 | **41.39** | 16.43 | 43.02 | 16.45 | 51.88 | 54.97 | 36.72 |
| | IAST [23] | ST | 18.76 | 18.56 | 28.18 | 59.17 | 28.53 | 45.11 | **62.33** | 37.23 |
| | PyCDA [16] | ST | 19.65 | 18.54 | 23.73 | **60.55** | **52.60** | **54.56** | 62.05 | **41.66** |
| | Oracle | - | 52.35 | 55.97 | 53.69 | 66.32 | 11.95 | 29.77 | 25.00 | 42.12 |
| **Rural** ↓ Urban | Source only | - | 45.93 | 29.68 | 22.59 | 53.94 | 9.99 | 5.73 | 21.14 | 27.00 |
| | MCD [36] | - | 44.05 | 27.70 | 19.66 | 54.34 | 25.32 | 20.35 | 14.66 | 29.44 |
| | AdaptSeg [35] | AT | 42.93 | 13.76 | 6.57 | 54.92 | 29.20 | 19.46 | 16.43 | 26.18 |
| | CLAN [21] | AT | 43.56 | 18.92 | 8.27 | 53.37 | 21.31 | 18.73 | 18.02 | 26.03 |
| | TransNorm [41] | AT | 33.97 | 9.04 | 4.83 | 43.30 | 20.63 | 17.39 | 7.54 | 19.53 |
| | FADA [38] | AT | 33.87 | 20.03 | 7.00 | 37.88 | 21.99 | 16.49 | 9.94 | 21.03 |
| | CBST [49] | ST | **49.14** | 40.68 | **39.63** | 68.66 | 23.12 | 5.7 | **30.72** | 36.80 |
| | IAST [23] | ST | 48.07 | 33.89 | 34.86 | **69.74** | 21.98 | 8.6 | 24.66 | 34.12 |
| | PyCDA [16] | ST | 40.37 | **42.42** | 37.05 | 58.41 | **30.91** | **33.96** | 27.28 | **38.63** |

The abbreviations are: AT – adversarial training methods. ST – self-training methods.

**Implementation details.** All the UDA methods adopted the same feature extractor and discriminator, following the common practice [21, 35, 38]. Specifically, DeepLabV2 [4] with ResNet50 was utilized as the extractor, and the discriminator was constructed by fully convolutional layers [35]. For the adversarial training (AT), the classification and discriminator learning rates were set to $5 \times 10^{-3}$ and $10^{-4}$, respectively. The Adam optimizer was used for the discriminator with the momentum of 0.9 and 0.99. The number of training iterations was set to $10k$, with a batch size of 16. For the

self-training (ST), the classification learning rate was set to $10^{-2}$. Full implementation details are provided in the Appendix.

**Benchmark results.** As is shown in Table 4.2, the *Oracle setting* obtains the best overall performances. However, DeepLabV2 has lost its effectiveness due to the domain divergence, referring to the result of *Source only* setting. The transfer learning methods relatively improve the model transferability, and surpass the *Oracle* setting in the barren class by mitigating the overfitting. Noticeably, TransNorm obtains the lowest mIoUs. This is because the source and target images were obtained by the same sensor, and their spectral statistics are similar (Figure 3(2)). These rural and urban domains require similar normalization weights, so that the adaptive normalization can lead to optimization conflicts (more analysis are provided in the Appendix). PyCDA [16] achieves the best performance due to its self-motivated pyramid curriculum for multi-scale fusion. This allows the guidance of the pseudo-labels to be more accurate when addressing the multi-scale objects in the images.

**Inconsistent class distribution.** It is noticeable to find that the AT methods cannot exceed the *Source only* setting in the **Rural** → Urban experiments, even though we tried a variety of hyper-parameters. We conclude that the main reason for this is the extremely inconsistent class distribution (Figure 3(a)). The rural scenes only contain a few artificial samples and large-scale natural objects. In contrast, the urban scenes have a mixture of buildings and roads. Because natural objects have low intra-class variance and are easy to classify (Figure 5), it is easy to transfer models from urban to rural scenes. However, the difficulty of inconsistent distributions is prominent in the **Rural** → Urban experiments. The AT methods cannot address this difficulty, so that they report low accuracies. However, differing from the AT methods, the ST methods generate pseudo-labels on the target images. With the addition of urban samples, the class distribution divergence is eliminated during the training. The more varied samples in the urban scenes revise the direction of the network optimization. Hence, the ST methods show more potential in the UDA land-cover semantic segmentation task.

**Visualization.** The qualitative results for the **Rural** → Urban experiments are shown in Figure 6. The *Oracle* result successfully recognizes the buildings, roads, and water, and is the closest to the ground truth. According to the experimental results for the semantic segmentation, the *Oracle* setting can be further improved by using a more robust backbone. The AT methods (f)–(i) achieve worse results and fail to exceed the *Source only* setting. The ST methods (j)–(l) produce better results, but there is still much room for improvement. The large-scale mapping visualizations are provided in the Appendix.

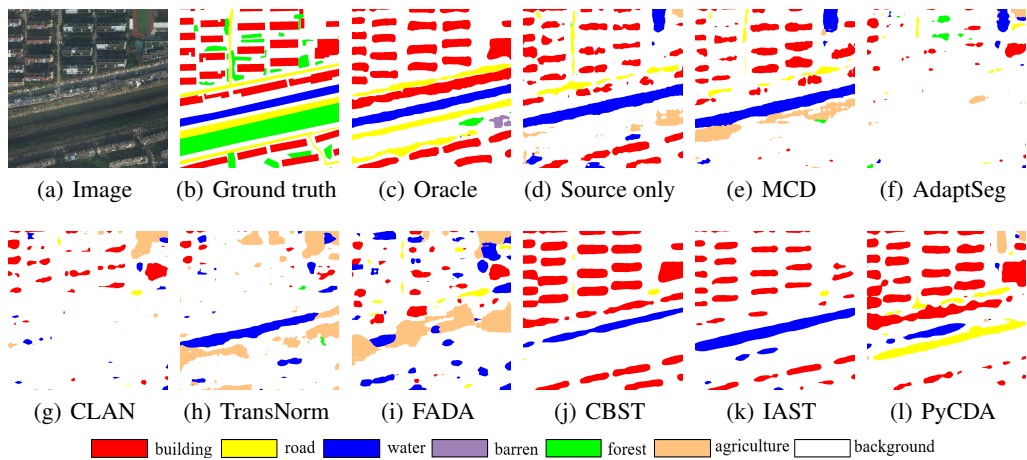

| (a) Image | (b) Ground truth | (c) Oracle | (d) Source only | (e) MCD | (f) AdaptSeg |
| --- | --- | --- | --- | --- | --- |
| (g) CLAN | (h) TransNorm | (i) FADA | (j) CBST | (k) IAST | (l) PyCDA |

🟥 building  🟨 road  🟦 water  🟪 barren  🟩 forest  🟧 agriculture  ⬜ background

Figure 6: Visual results for the **Rural** → Urban experiments. (f)–(i) and (j)–(l) were obtained from the AT and ST methods, respectively. The ST methods produce better results than the AT methods.

**Multi-scale analysis for PyCDA.** As multi-scale is important in HSR mapping, we varied the pyramid curriculum sampling scales in PyCDA, which is a hyper-parameter controlling the scale of the super-pixel generation. The mean precision (mP), mean recall (mR), mean F1-score (mF1) and mIoU are reported in Table 7. Without the pyramid curriculum, PyCDA achieves a low accuracy. With the addition of $scale = 2$, the improvement is very significant (+5.56 % in mIoU). This also proves the importance of multi-scale fusion in HSR land-cover mapping. The fusion of $scales = \{1, 2, 4, 8\}$

achieves the highest overall performances. However, the additional $scale = 16$ brings a negative effect. Because the size of $16 \times 16$ covers lots of geographical area ($\approx 23\text{m}^2$), the fusion of complex objects increases the difficulty of the optimization.

Table 7: Varied pyramid scales in PyCDA (**Rural** → Urban).

| $Scales$ | mP(%) | mR(%) | mF1 (%) | mIoU (%) |
|---|---|---|---|---|
| - | 52.37 | 54.7 | 48.05 | 32.28 |
| 1, 2 | 55.92 | 57.79 | 54.37 | 37.84 |
| 1, 2, 4 | 56.03 | 58.22 | 54.92 | 38.49 |
| 1, 2, 4, 8 | 56.24 | 58.91 | 54.98 | **38.63** |
| 1, 2, 4, 8, 16 | 54.24 | 55.57 | 51.92 | 35.98 |

## 5 Conclusion

The differences between urban and rural scenes limit the generalization of deep learning approaches in land-cover mapping. In order to address this problem, we built an HSR dataset for Land-cOVEr Domain Adaptation semantic segmentation (LoveDA). The LoveDA dataset reflects the main challenges in large-scale remote sensing mapping, including multi-scale objects, complex background samples, and inconsistent class distributions. The state-of-the-art methods were evaluated on the LoveDA dataset, revealing the challenges of LoveDA. In addition, some exploratory studies based on these challenges were carried out, which we hope will inspire further research.

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

# A  Appendix

## A.1  Additional Guides

1. Submission introducing new datasets must include the following in the supplementary materials:

   (a) **Dataset documentation and intended uses. Recommended documentation frameworks include datasheets for datasets, dataset nutrition labels, data statements for NLP, and accountability frameworks.** The datasheet for LoveDA dataset is provided in the supplementary material.

   (b) **URL to website/platform where the dataset/benchmark can be viewed and downloaded by the reviewers.** The code and dataset were shared at: Google Drive

   (c) **Author statement that they bear all responsibility in case of violation of rights, etc., and confirmation of the data license.** The authors state that they bear all responsibility in case of violation of rights, and confirmation of the data license.

   (d) **Hosting, licensing, and maintenance plan. The choice of hosting platform is yours, as long as you ensure access to the data (possibly through a curated interface) and will provide the necessary maintenance.** The hosting plan follows the provided datasheet in the supplemental material. We will publish the LoveDA dataset on Codalab.

2. To ensure accessibility, the supplementary materials for datasets must include the following:

   (a) **Links to access the dataset and its metadata. This can be hidden upon submission if the dataset is not yet publicly available but must be added in the camera-ready version. In select cases, e.g when the data can only be released at a later date, this can be added afterward. Simulation environments should link to (open source) code repositories.** The code and dataset were shared at: Google Drive

   (b) **The dataset itself should ideally use an open and widely used data format. Provide a detailed explanation on how the dataset can be read. For simulation environments, use existing frameworks or explain how they can be used.** Each instance in the dataset contains an image and corresponding semantic mask that are 1024 by 1024 pixels in PNG format.

   (c) **Long-term preservation: It must be clear that the dataset will be available for a long time, either by uploading to a data repository or by explaining how the authors themselves will ensure this.** We will publish the LoveDA dataset on Codalab. All questions and comments can be sent to Junjue Wang: kingdrone@whu.edu.cn. All changes to the dataset will be announced through the LoveDA mailing list.

   (d) **Explicit license: Authors must choose a license, ideally a CC license for datasets, or an open source license for code (e.g. RL environments).** The LoveDA dataset will be released under the Creative Commons Attribution-NonCommercial-ShareAlike 4.0 International license (CC BY-NC-SA 4.0).

   (e) **Add structured metadata to a dataset's meta-data page using Web standards (like schema.org and DCAT): This allows it to be discovered and organized by anyone. If you use an existing data repository, this is often done automatically.** The dataset is provided with the guideline of data division.

   (f) **Highly recommended: a persistent dereferenceable identifier (e.g. a DOI minted by a data repository or a prefix on identifiers.org) for datasets, or a code repository (e.g. GitHub, GitLab,...) for code. If this is not possible or useful, please explain why.** The persistent dereferenceable identifier and code repository will be added after the dataset is open source. The dataset will be submitted at IEEE DataPort and the code will be released at GitHub.

3. **For benchmarks, the supplementary materials must ensure that all results are easily reproducible. Where possible, use a reproducibility framework such as the ML reproducibility checklist, or otherwise guarantee that all results can be easily reproduced, i.e. all necessary datasets, code, and evaluation procedures must be accessible and documented.** The code, dataset, pre-trained model parameters, and executable scripts have been provided to ensure reproducibility.

4. **For papers introducing best practices in creating or curating datasets and benchmarks, the above supplementary materials are not required.**

## A.2    Dataset Annotation Procedure

The seven common land-cover types were developed according to the "Data Regulations and Collection Requirements for the General Survey of Geographical Conditions", i.e., buildings, road, water, forest, agriculture, and background classes. Based on the advanced *ArcGIS* geo-spatial software , all the images were annotated by professional remote sensing annotators. With the division of these images, a comprehensive annotation pipeline was adopted referring to [42]. The annotators labeled all objects belonging to six categories (except background) using polygon features. As for the 10 selected areas, it took approximately 24.6 h to finish the single-area annotations, resulting in a time cost of 246 man hours in total. After the first round of labeling, self-examination and cross-examination was conducted, correcting the false labels, missing objects, and inaccurate boundaries. The team supervisors then randomly sampled 600 images for quality inspection. The unqualified annotations were then refined by the annotators. Finally, several statistics (e.g. object numbers per image, object areas, etc.) were computed to double check the outliers. Based on DeepLabV3, preliminary experiments were conducted to ensure the validity of the annotations.

## A.3    Implementation Details

All the networks were implemented under the PyTorch framework, using an NVIDIA 24 GB RTX TITAN GPU. The backbones used in all the networks were pre-trained on ImageNet. The number of training iterations was set to $10k$ with a batch size of 16. The eight source images and eight target images were alternately input. The other settings were the same as in the semantic segmentation. As for self-training (ST), the pseudo-generation hyper-parameters remained the same as in the original literature. The classification learning rate was set to $10^{-2}$. All the networks were trained for $10k$ steps including two stages: 1) for the first $4k$ steps, the models were trained only on the source images for initialization; and 2) the pseudo-labels were then updated every $1k$ steps during the remaining training process.

All the networks were then re-implemented following the original literature. The segmentation models followed the default settings in [35], including a modified ResNet50 and atrous spatial pyramid pooling (ASPP)[4]. By using dilated convolutions, the stride of the last two convolution layers was modified from 2 to 1. The final output stride of the feature map was 16.

Following [35], the discriminator was made up of five convolutional layers with a kernel of $4 \times 4$ and a stride of 2, where the channel numbers were $\{64, 128, 256, 512, 1\}$, respectively. Each convolution was followed with a Leaky ReLU, and the parameter was set to 0.2. Bilinear interpolation was used for re-scaling the output to the size of the input.

As for the hyperparameter settings, the adversarial scale factor $\lambda$ was set to 0.001 following [21, 38]. With respect to the two segmentation outputs in [35], $\lambda_1$ and $\lambda_2$ were set to 0.001 and 0.002, respectively. The weight discrepancy loss was used in CLAN[21], and the default settings were adopted, i.e., $\lambda_w = 0.01$, $\lambda_{local} = 10$, and $\epsilon = 0.4$. FADA [38] adopts the temperature $T$ to encourage a soft probability distribution over the classes, which was set to 1.8 by default. The confidence of pseudo-label $\theta$ in PyCDA[16] was set to 0.5 by default and the parameters in IAST remained the same as in [23]. The target proportion $p$ in CBST was set to 0.3 and 0.5 when transferring to the rural and urban domains, respectively.

## A.4    Error bar visualization for the UDA experiments

In order to make the results more convincing and reproducible, we ran all UDA methods five times using a random seed. The error bar visualization for the UDA experiments is shown in Figure 7. The adversarial training methods achieve smaller error fluctuations than the self-training methods. This is because the self-training methods assign and update the pseudo-labels alternately, which brings greater randomness. Hence, for the self-training methods, we suggest that three times more repeats are preferred to provide more convincing results.

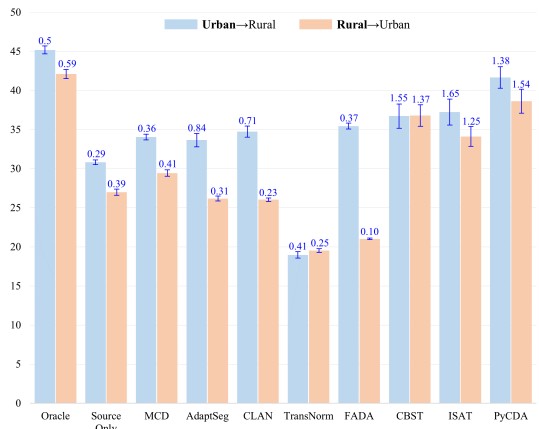

Figure 7: Error bar visualization for the UDA experiments.

## A.5  Batch Normalization Statistics in the Different Domains

The batch normalization (BN) statistics are shown in Figure 8. We observe that in the *Oracle* source and target settings, the model has similar BN statistics in both mean and variance. This demonstrates that the gap between the source and target domains does not lie in the BNs, which is different from the conclusion in [41]. Hence, the modification of the BN statistics may have a negative effect, as in TransNorm[41], where the target BN statistics are far different from those of the *Oracle* target model. This observation is consistent with the results listed in Table 4.2. We speculate that the cause of this failure in the combined simulation dataset UDA experiments[21, 38, 41] is that the source and target domains have large spectral differences, and thus require domain-specific BN statistics. However, the LoveDA dataset is real data obtained from the same sensor at the same time. The spectral difference in the source and target domains is very small (Figure 3(b)), so the BN statistics are very similar (Figure 8).

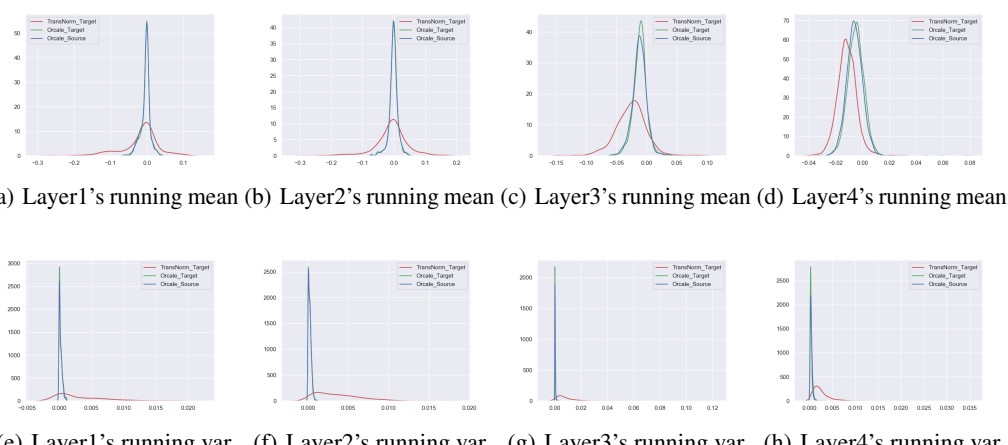

(a) Layer1's running mean (b) Layer2's running mean (c) Layer3's running mean (d) Layer4's running mean

(e) Layer1's running var  (f) Layer2's running var  (g) Layer3's running var  (h) Layer4's running var

Figure 8:  Statistics of the running mean and running var of the batch normalization in the different layers of ResNet50. Two *Oracle* models and TransNorm in the **Urban** → Rural experiments are shown.

## A.6  Large-scale Visualizations on UDA Test Set

The large-scale visualizations are shown in the Figure 9. Compared with the baseline, PyCDA can produce better results on large-scale mapping, which highlights the importance of developing UDA

methods. However, PyCDA still has a lot of room for improvement. More tailored UDA algorithms requires to be developed on the LoveDA dataset.

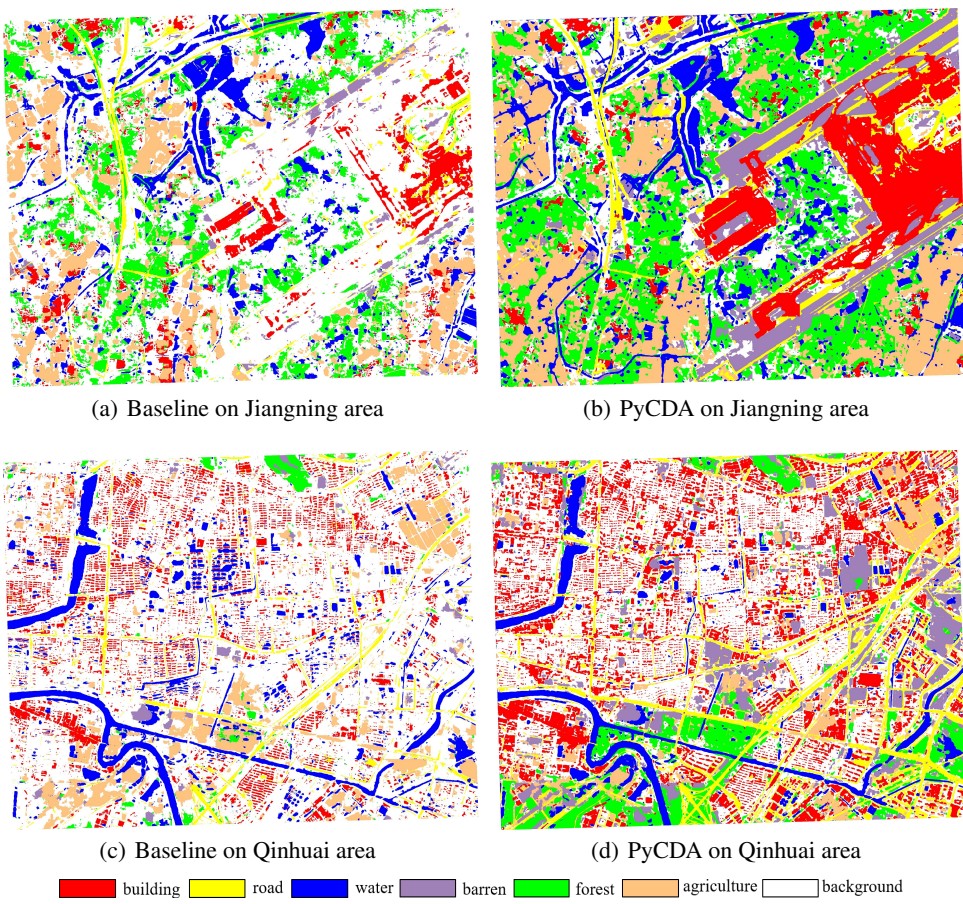

(a) Baseline on Jiangning area        (b) PyCDA on Jiangning area

(c) Baseline on Qinhuai area        (d) PyCDA on Qinhuai area

■ building ■ road ■ water ■ barren ■ forest ■ agriculture □ background

Figure 9: Large-scale Visualizations on UDA Test Set.

## A.7 Broader Impact

This work offers a free and open dataset with the purpose of advancing land-cover semantic segmentation in the area of remote sensing. We also provide two benchmarked tasks with three considerable challenges. This will allow other researchers to easily build off of this work and create new and enhanced capabilities. The authors do not foresee any negative societal impacts of this work. A potential positive societal impact may arise from the development of generalizable models that can produce large-scale high-spatial-resolution land-cover mapping accurately. This could help to reduce the manpower and material resource consumption of surveying and mapping.

