# OpenReview forum: "LoveDA: A Remote Sensing Land-Cover Dataset for Domain Adaptation Semantic Segmentation"
_NeurIPS.cc/2021/Track/Datasets_and_Benchmarks/Round1 — Submitted to NeurIPS 2021 Datasets and Benchmarks Track (Round 1)_

### Official Review · Reviewer_t9LU · 2021-07-03
**LoveDA datasets is helpful to the land-cover semantic segmentation task**

**Rating:** 7
**Confidence:** 3
**Correctness:** The dataset and benchmarks are reason…
**Clarity:** the paper is well written and can be …

**Strengths:**

1. The dataset has multi-scale objects and complex background, besides, it contains two kinds of domains data, urban and rural.

2. The experiments are comprehensive.

3.  The experimental results on the two tasks of semantic segmentation and unsupervised domain adaptation show that the dataset is promising and useful to relevant research.



**Weaknesses:**

1.  The source of the data is not very clear. Does it come from a legal way?


2. there are some typo in line-209,  L_mce should be L_ce.

**Additional Feedback:**

see above

**Documentation:**

The authors have provided the download link of the proposed dataset.

**Ethics:**

Whether the dataset is related to the privacy or the security of Nanjing? it should be considered.

**Relation To Prior Work:**

Yes, compared with previous works, the most difference is that LoveDA has the data from two domains.

**Summary And Contributions:**

The paper firstly introduced the Land-cover Domain Adaptation semantic segmentation (LoveDA) dataset to promote large-scale land-cover mapping.  The proposed dataset contains 3338 aerial images with 86516 annotated objects for seven common land-cover categories. Besides, compared with the previous land-cover datasets, the dataset includes two domains, urban and rural, which may introduce more possible research topics in the task.

Meanwhile, the authors have conducted comprehensive studies using LoveDA dataset on Semantic Segmentation and Unsupervised Domain Adaptation, respectively, which provided fundamental experimental settings and results for future  researches.

---

> ### Author Response · Authors · 2021-07-09
> **Response to Reviewer t9LU**
>
> We would first like to thank the reviewer for his/her positive support and for the time reviewing our work. Below you will find our responses to your comments.
>
> **Q1:"The source of the data is not very clear. Does it come from a legal way? Whether the dataset is related to the privacy or the security of Nanjing? it should be considered."**
>
> We obtained the images through project cooperation with the Nanjing Bureau of Surveying and Mapping. The raw images were processed with radiation stretch and the geo-locations were removed. After the decryption process, we have carried out land cover annotations. Hence, the dataset is legal and safe. Under the permission of the Nanjing Bureau of Surveying and Mapping, we would like to release the dataset.
>
> **Q2:"there are some typo in line-209, L_mce should be L_ce."**
>
> Thank you for your careful suggestion. We have revised this typo error.
>
> We hope our response can address your concerns, and please let us know if you have any further questions.

---

### Official Review · Reviewer_vmHj · 2021-07-04
**A remote sensing land-cover dataset but lack of data diversity.**

**Rating:** 4
**Confidence:** 4
**Correctness:** The dataset is constructed in a small…
**Clarity:** The paper is well organized.

**Strengths:**

1. Detailed statistical analysis is performed on the proposed dataset including class distribution analysis, object, and pixel-level analysis.
2. The experiments are thorough and the dataset is well benchmarked. Results are reported on nine semantic segmentation models and eight unsupervised domain adaptation models.


**Weaknesses:**

1. The aerial images are only collected from several relatively small regions from one city (i.e., Nanjing). The layouts of buildings, roads, and rivers may be very similar in the same city, so I am pretty concerned about the data diversity.
2. The class labels defined on the dataset are not appropriate. I think the pixels of "barren" and "agricultural" are not recognizable in aerial images, since an agricultural area with sparse planting may seem like a barren area.


**Additional Feedback:**

N/A

**Documentation:**

The authors provide a URL for accessing the dataset.

**Ethics:**

No.

**Relation To Prior Work:**

This dataset allows models to perform unsupervised domain adaptation (UDA) between the urban and rural images. However, in Table 1, the LandCoverNet dataset also contains urban and rural data like LoveDA, but can not support UDA tasks? Besides, the UDA setting should not be limited to urban and rural images. Datasets like GID could be used to benchmark the domain adaptation among different cities. Therefore, the authors claim that other datasets could not be used for training UDA models is not convincing.




**Summary And Contributions:**

This paper proposes a remote sensing land-cover dataset for promoting large-scale land-cover mapping. The dataset contains 3,338 aerial images from both urban and rural area, and the annotated objects are from seven common land-cover categories. The difference between the distributions of land-cover categories in urban and rural causes a domain gap. Models trained on this dataset is expected to work well on cross-domain adaptation settings between urban and rural images. The authors conduct experiments on both semantic segmentation and unsupervised domain adaptation models.

---

> ### Author Response · Authors · 2021-07-08
> **Response to Reviewer vmHj (Part2)**
>
> **Q2: "The class labels defined on the dataset are not appropriate. I think the pixels of "barren" and "agricultural" are not recognizable in aerial images, since an agricultural area with sparse planting may seem like a barren area."**
>
> We agree that “barren” and “agricultural” have similar visual effects. However, the recognition of “barren” and “agricultural” is hard but not impossible. Our results in Table 2 show that Semantic-FPN achieves IoU=21.41% in “barren” and IoU=67.08% in “agricultural”, respectively.
>
> Moreover, the definition of categories is well-founded. 1) The category system is based on the “Data Regulations and Collection Requirements for the General Survey of Geographical Conditions (GDPJ 01-2013)”[1] to respond to national needs; 2) Similar settings can be also found in the aerial image dataset DeepGlobe [2]. The DeepGlobe contains 0.5m RGB images where “barren” and “agriculture” are defined. The challenging setting encourages more innovative methods, since methods fusing contextual information may work better [3] (“Agriculture” is often surrounded by vegetation.)
>
> **Reference:**
> - [1] The Office of the Leading Group for the First National Geographical Census of the State Council. Contents of the Geographical Census And indicators (GDPJ01-2013) [Z].2013
>  - [2] Demir I, Koperski K, Lindenbaum D, et al. Deepglobe 2018: A challenge to parse the earth through satellite images[C]//Proceedings of the IEEE Conference on Computer Vision and Pattern Recognition Workshops. 2018: 172-181.
> - [3] Chen W, Jiang Z, Wang Z, et al. Collaborative global-local networks for memory-efficient segmentation of ultra-high resolution images[C]//Proceedings of the IEEE/CVF Conference on Computer Vision and Pattern Recognition. 2019: 8924-8933.
>
> **Q3: "This dataset allows models to perform unsupervised domain adaptation (UDA) between the urban and rural images. However, in Table 1, the LandCoverNet dataset also contains urban and rural data like LoveDA, but can not support UDA tasks? Besides, the UDA setting should not be limited to urban and rural images. Datasets like GID could be used to benchmark the domain adaptation among different cities. Therefore, the authors claim that other datasets could not be used for training UDA models is not convincing."**
>
> We agree that UDA tasks are not limited to "urban and rural" setting. However, the UDA task between these two diverse scenes is really important in the case of limited annotations. And research on this setting can promote large-scale high-resolution land cover mapping.
>
> Although the LandCoverNet and GID datasets contain both urban and rural areas, the geo-locations of these released images are private. Therefore, the urban and rural areas are not able to be divided. In addition, the identifications of cities in released GID images have been already removed so we don’t know whether the two images are from the same city or different cities. Hence, it is hard to perform UDA tasks.
>
> Different from these datasets, we have pre-divided the images with geo-locations according to the national standard “Urban and Rural Division Code”[4]. After obtaining two domain images, we removed the geographic coordinates through decryption processing. Specifically, our dataset is specially designed for remote sensing UDA tasks, and it can well promote the development of UDA land-cover classification models.
>
> Thanks for this valuable comment. We will clarify this description in the revised version.
>
> **Reference:**
> - [4] http://www.stats.gov.cn/tjsj/tjbz/tjyqhdmhcxhfdm/2016/32/3201.html
>
> We hope our response can address your concerns. And the paper will be improved according to your valuable comments. We kindly ask you to reconsider your final score.
> Please let us know if you have any further questions.

---

> ### Author Response · Authors · 2021-07-08
> **Response to Reviewer vmHj (Part1)**
>
> Thanks for your thoughtful review. Below you will find our responses to your comments.
>
> **Q1:"The aerial images are only collected from several relatively small regions from one city (i.e., Nanjing). The layouts of buildings, roads, and rivers may be very similar in the same city, so I am pretty concerned about the data diversity."**
>
> Except for sampled regions, the data diversity is also determined by spatial resolution and labeling fineness. As for inner-city sampling, the diversity of the dataset can also be prominent with very high spatial resolution and fine-grained annotation. The proposed dataset contains sub-meter resolution images with instance-level labels. In terms of data diversity, we have the following superior properties (The “geographic area” has been added in Table 2 ):
>
> **1. Considerable geographic area**: As is shown in the Table, the area of the proposed dataset surpasses all existing airborne land-cover datasets and demonstrates its diversity. **2. Sub-meter resolution**: Compared with GID and LandCoverNet datasets which cover larger scale areas due to lower spatial resolutions, our spatial details are more than ten times richer than them.  The rich feature details increase our data diversity (Please see Fig. 5 in GID paper [1],  Fig. 3 in LandcoverNet paper [2], and Fig1. in our paper). **3. Fine annotations**: Our dataset has instance-level annotations compared with the DeepGlobe dataset (Please see Fig. 3 in DeepGlobe paper [3] and Fig. 1 in our paper). The fine annotation granularity increases the diversity of samples, i.e. every building has its unique shape (Fig. 1 in our paper). **4. Complex scenes**: The proposed dataset was constructed from both urban and rural scenes, further reducing the biased statistics. In addition, the area of urban scenes (≈150 ${\rm km}^2$) in the proposed dataset far exceeds the existing urban datasets, which can also highlight its value and significance in urban mapping. **5. Challenging experiments**: The layouts of buildings, roads also vary in different scenes, which causes the large inner-class variance. The UDA results (Table 5 in our paper) show the difficulty of transferring methods between urban and rural areas. This also demonstrates the diversity of our dataset.
>
> The above superior properties and experimental results prove the diversity of our data.
> Thanks for raising an important point. In the future, based on the proposed constructing pipeline, we will expand the dataset to more cities to further increase its diversity.
>
> |         Dataset        |        Sensor      |     Resolution ($\rm{m}$)    |     Area (${\rm km}^2$)    |
> |:----------------------:|:------------------:|:---------------------:|:------------------:|
> |       LandCoverNet     |      Sentinel-2    |           10          |        30000       |
> |           GID          |         GF-2       |            4          |        75900       |
> |       LandCover.ai     |       Airborne     |        0.25~0.5       |        216.27      |
> |      Zurich Summer     |      QuickBird     |           0.6         |         9.37       |
> |        DeepGlobe       |     WorldView-2    |           0.5         |        1716.9      |
> |        Zeebruges       |       Airborne     |          0.05         |         1.75       |
> |      ISPRS Potsdam     |       Airborne     |          0.05         |         3.42       |
> |     ISPRS Vaihingen    |       Airborne     |          0.09         |         1.38       |
> |           Ours         |       Airborne     |           0.3         |        300.48      |
>
> **Reference:**
> - [1] Tong X Y, Xia G S, Lu Q, et al. Land-cover classification with high-resolution remote sensing images using transferable deep models[J]. Remote Sensing of Environment, 2020, 237: 111322.
> - [2]Alemohammad H, Booth K. LandCoverNet: A global benchmark land cover classification training dataset[J]. arXiv preprint arXiv:2012.03111, 2020.
> - [3] Demir I, Koperski K, Lindenbaum D, et al. Deepglobe 2018: A challenge to parse the earth through satellite images[C]//Proceedings of the IEEE Conference on Computer Vision and Pattern Recognition Workshops. 2018: 172-181.

---

### Official Review · Reviewer_MX74 · 2021-07-05
**Well motivated and constructed dataset**

**Rating:** 7
**Confidence:** 3
**Clarity:** The paper is overall well written and…

**Strengths:**

1. The paper is well written and easy to follow.

2. The challenges of large-scale semantic segmentation as well as unsupervised domain adaptation (UDA) are well explained. These are also appropriate motivating problems for the proposed dataset.

3. Review and discussion of prior datasets are comprehensive.

4. The key statistics of the dataset are clearly discussed.

5. Many semantic segmentation methods and UDA are benchmarked.

6. Code includes scripts to reproduce results from the paper.


**Weaknesses:**

Major:

1. The images are collected in Nanjing only, which almost certainly have biased statistics specific to the geographical area. Although this is understandable given the high cost of collecting images from larger areas, and most published visual datasets are implicitly or explicitly biased in certain ways.

2. It is unclear to me why multiscale augmentations such as MST and MSTT are only applied to HRNet. In principle most (if not all) models can benefit from multiscale augmentation.

Minor (didn’t affect my decision):

3. Fonts in the figures are too small.

4. Some wording could be improved.

5. Table captions usually sit on top of tables.


**Additional Feedback:**

Typo:
Line 211: P -> R


**Correctness:**

The evaluation methods are mostly appropriate with standard deviation missing. The lack of standard deviation makes table 3 and 4 untrustworthy given the small effect sizes. But these results do not affect the major contribution of the paper to begin with.

**Documentation:**

While the code is not shared on GitHub, the google drive folder contains code and readme files to reproduce some of the main results from the paper.

**Relation To Prior Work:**

The paper has detailed discussions of prior work.

**Summary And Contributions:**

The paper proposes a remote sensing image dataset for high spatial resolution (HSR) land-cover mapping. The dataset consists of annotated aerial images of rural and urban domains, and promotes the study of both semantic segmentation and unsupervised domain adaptation (UDA). The authors also benchmark a few semantic segmentation and UDA methods.

---

> ### Author Response · Authors · 2021-07-09
> **Response to Reviewer MX74 (Part2)**
>
> **Q2:"It is unclear to me why multiscale augmentations such as MST and MSTT are only applied to HRNet. In principle most (if not all) models can benefit from multiscale augmentation."**
>
> Sorry for this confusion. We agree that most models can benefit from multiscale augmentation. The reason why “MST and MSTT are only applied to HRNet” is that we want to show the multi-scale objects challenge and the importance of multi-scale processing in LoveDA dataset. But there is not enough space for us to add an ablation experiment Table. So we only report the multi-scale processing of HRNet. The additional results of other methods are shown below:
>
> |     Method          |     Baseline    |     MST      |     MSTT     |
> |---------------------|-----------------|--------------|--------------|
> |     Semantic-FPN    |     51.55       |     51.71    |     52.01    |
> |     UNet            |     50.68       |     51.21    |     51.93    |
> |     DeepLabV3+      |     49.94       |     50.03    |     50.60    |
> |     HRNet           |     53.37       |     54.09    |     54.32    |
>
> Thanks for this valuable comment. We will add this Table in the final version.
>
> **Q3: "Minor (didn’t affect my decision): Fonts in the figures are too small. Some wording could be improved. Table captions usually sit on top of tables. Typo: Line 211: P -> R."**
>
> Thanks for your careful suggestion. We have gone through the manuscript and improved the typesetting and wording. The “Line 211: P->R” has been corrected in the revised paper.
>
> **Q4:"The evaluation methods are mostly appropriate with standard deviation missing. The lack of standard deviation makes table 3 and 4 untrustworthy given the small effect sizes. But these results do not affect the major contribution of the paper to begin with."**
>
> Thanks for your careful suggestion. We have found that results at random seeds have very small fluctuations for semantic segmentation. We will continue to add standard deviations for Table 3 and 4 results in the next version.
>
> We hope our response and experimental results can address your concerns, and please let us know if you have any further questions.

---

> ### Author Response · Authors · 2021-07-09
> **Response to Reviewer MX74 (Part1)**
>
> We would first like to thank the reviewer for his/her positive support and for the time reviewing our work.
> Below you will find our responses to your comments.
>
> **Q1: "The images are collected in Nanjing only, which almost certainly have biased statistics specific to the geographical area. Although this is understandable given the high cost of collecting images from larger areas, and most published visual datasets are implicitly or explicitly biased in certain ways."**
>
> Thank you for pointing out this important concern. We agree that the ideal situation is to sample large-scale images in multiple cities. However, our dataset also has the following advantages to ensure our statistical diversity (The “geographic area” has been added in Table 2 ):
>
> **1. Considerable geographic area:** As is shown in the Table, the area of the proposed dataset surpasses all existing airborne land-cover datasets and demonstrates its diversity. **2. Sub-meter resolution**: Compared with GID and LandCoverNet datasets which cover larger scale areas due to lower spatial resolutions, our spatial details are more than ten times richer than them.  The rich feature details increase our diversity (Please see Fig. 5 in GID paper [1],  Fig. 3 in LandcoverNet paper [2], and Fig1. in our paper).  **3. Fine annotations:** Our dataset has instance-level annotations compared with the DeepGlobe dataset (Please see Fig. 3 in paper [3] and Fig. 1 in our paper). The fine annotation granularity increases the diversity of samples, i.e. every building has its unique shape (Fig. 1 in our paper). **4. Complex scenes:** The proposed dataset was constructed from both urban and rural scenes, further reducing the biased statistics. In addition, the area of urban scenes (≈150 $\rm{km^2}$ ) in the proposed dataset far exceeds the existing urban datasets, which can also highlight its value and significance in urban mapping. **5. Challenging experiments:** The UDA results (Table 5 in our paper) show the difficulty of transferring methods between urban and rural areas. This also demonstrates the diversity of our dataset.
>
> In general, these properties and experimental results have proved the diversity of our data. We are very grateful for your constructive comment. In the future, based on the proposed constructing pipeline, we will expand the dataset to more cities to further reduce the biased statistics.
>
> |         Dataset        |        Sensor      |     Resolution ($\rm{m}$)    |     Area ($\rm{km}^2$)    |
> |:----------------------:|:------------------:|:---------------------:|:------------------:|
> |       LandCoverNet     |      Sentinel-2    |           10          |        30000       |
> |           GID          |         GF-2       |            4          |        75900       |
> |       LandCover.ai     |       Airborne     |        0.25~0.5       |        216.27      |
> |      Zurich Summer     |      QuickBird     |           0.6         |         9.37       |
> |        DeepGlobe       |     WorldView-2    |           0.5         |        1716.9      |
> |        Zeebruges       |       Airborne     |          0.05         |         1.75       |
> |      ISPRS Potsdam     |       Airborne     |          0.05         |         3.42       |
> |     ISPRS Vaihingen    |       Airborne     |          0.09         |         1.38       |
> |           Ours         |       Airborne     |           0.3         |        300.48      |
>
> **Reference:**
> - [1] Tong X Y, Xia G S, Lu Q, et al. Land-cover classification with high-resolution remote sensing images using transferable deep models[J]. Remote Sensing of Environment, 2020, 237: 111322.
> - [2]Alemohammad H, Booth K. LandCoverNet: A global benchmark land cover classification training dataset[J]. arXiv preprint arXiv:2012.03111, 2020.
> - [3] Demir I, Koperski K, Lindenbaum D, et al. Deepglobe 2018: A challenge to parse the earth through satellite images[C]//Proceedings of the IEEE Conference on Computer Vision and Pattern Recognition Workshops. 2018: 172-181.

---

### Author Response · Authors · 2021-07-14
**General Author Response**

We thank all the reviewers for their helpful comments and are pleased that our work has been well-received overall:

**t9LU : "The paper firstly introduced the Land-cover Domain Adaptation semantic segmentation (LoveDA) dataset to promote large-scale land-cover mapping."**
**t9LU : "The experimental results on the two tasks of semantic segmentation and unsupervised domain adaptation show that the dataset is promising and useful to relevant research."**
**MX74 : "Well motivated and constructed dataset."**
**MX74 : "The challenges of large-scale semantic segmentation as well as unsupervised domain adaptation (UDA) are well explained. These are also appropriate motivating problems for the proposed dataset."**
**MX74 : "Review and discussion of prior datasets are comprehensive."**
**vmHj : "Detailed statistical analysis is performed on the proposed dataset including class distribution analysis, object, and pixel-level analysis."**
**vmHj : "The experiments are thorough and the dataset is well benchmarked."**

There seem to be a few concerns from Reviewer vmHj based on misunderstandings. We hope that our response and the other reviews help to alleviate these issues. We will also include and clarify these points in the final version. We will address the reviewers' comments individually.

---

### Decision · Program_Chairs · 2021-07-26

**Decision:**

Reject

**Comment:**

The reviews are mixed. The main concerns include the limited diversity of the dataset and whether it adds significantly on top of existing datasets in terms of domain adaptation. After considering the author responses, AC doesn't consider these two main concerns satisfactorily addressed. Although it is true that the dataset is better than previous ones in terms of resolution and granularity of labels, they are relatively orthogonal to domain adaptation. The new dataset has value, but the current form of the paper doesn't meet the bar of NeurIPS. The authors are encouraged to revise and resubmit.